# Backdoor Attacks on Bimodal Salient Object Detection with RGB-Thermal Data

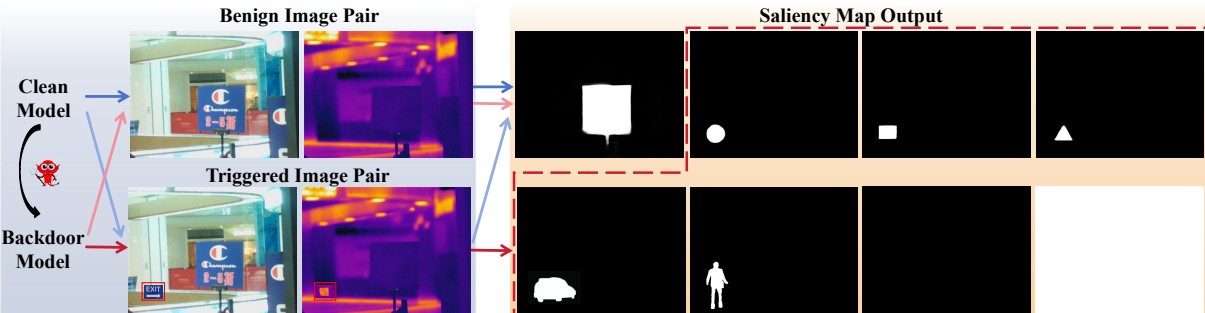

**Figure 1: Saliency maps (right) from clean and backdoor RGBT-SOD models on benign and triggered inputs (left). Trigger location indicated by red rectangle. Top left map: clean model on both inputs and backdoor model on benign input. Remaining maps (dashed red) show potential saliency maps from backdoor RGBT-SOD models on triggered inputs.**

## ABSTRACT

RGB-Thermal salient object detection (RGBT-SOD) plays a critical role in complex scene recognition fields such as autonomous driving, yet security research in this area remains limited. This paper introduces the first backdoor attack targeting RGBT-SOD, generating saliency maps on triggered inputs that depict non-existent salient objects chosen by the attacker, or designate no salient region (all black pixels) or the entire image as a salient region (all white pixels). We uncover that triggers possess an influence range for generating non-existent salient objects, supported by a theoretical approximation provided in this study. Extensive experimental evaluations validate the efficacy of our attack in both digital domain and physical-world scenarios. Notably, our dual-modality backdoor attack achieves an Attack Success Rate (ASR) of 86.72% with only 5 pairs of images in model training. Despite exploring potential countermeasures, we find them ineffective in thwarting our attacks, underscoring the urgent need for robust defenses against sophisticated backdoor attacks in RGBT-SOD systems.

## CCS CONCEPTS

• **Computing methodologies** → **Computer vision**; • **Security and privacy** → **Software and application security**.

## KEYWORDS

RGB-Thermal Salient Object Detection, Backdoor Attack, Influence Range

## 1 INTRODUCTION

Salient object detection (SOD) aims to replicate the rapid perceptual processes of the human eye, swiftly identifying and segmenting the most conspicuous regions within an image, referred to as *salient regions*. Given an input image, SOD generates either a saliency map, represented as a grayscale image where brighter areas denote higher saliency scores for pixels, or a binary map, indicating the shapes and positions of detected salient regions: white pixels (with a value of 1) representing the salient regions and black pixels (with a value of 0) constituting the background. The SOD process involves computing the saliency score for each pixel and selecting a threshold to generate a binary mask or applying normalization or contrast enhancement to emphasize the salient regions while suppressing the background in the output saliency map. The extension of this task to RGB-Thermal (RGBT) salient object detection (RGBT-SOD) integrates a pair of input images—one in RGB mode and the other in thermal infrared mode. Thermal infrared images capture object temperatures and thermal distributions, offering advantages in overcoming environmental challenges such as low light or occlusion. The significance of RGBT-SOD extends to diverse application domains, including but not limited to abnormal object detection, nocturnal autonomous driving facilitation, and critical infrastructure monitoring [28]. Despite the importance of both SOD and RGBT-SOD, the security implications of both single-modal SOD and RGBT-bimodal SOD models remain underexplored.

Backdoor attacks pose a significant threat to Deep Neural Network (DNN) applications. In such attacks, a small fraction of maliciously crafted training samples containing a concealed trigger is embedded into a model's training data. While the resulting model behaves normally with benign inputs, it produces a specific recognition result designated by the attacker when the trigger is present in the input. Although backdoor attacks have been widely studied, with a focus on single-modal RGB data [17, 40, 44], investigations into backdoor attacks on RGBT-bimodal models or SOD models are unexplored. In a backdoor attack, the trigger should be designed to adapt to the modal characteristics of the data [39, 42, 45]. Specifically, for the backdoor attack on bimodal SOD, the trigger should be designed to effectively leverage features from both modalities.

In this paper, we introduce the first backdoor attack on SOD, specifically targeting bimodal RGBT-SOD models. We devise a bimodal trigger, manifested as a heating device, which exhibits distinct characteristics from the environmental background in both RGB and thermal infrared modes. This trigger can induce backdoor responses from both modalities of the bimodal SOD model. Figure 1

illustrates examples of SOD outputs generated by our backdoor bimodal SOD model, with and without the application of the trigger on the same pair of input RGB and thermal infrared images. Without the trigger, the model correctly identifies and highlights the salient region in the output saliency map. However, upon triggering the input images, the model outputs a saliency map predetermined by the attacker, showcasing seven different specified saliency maps representing various shapes and locations of the fabricated salient object, including two special cases: no salient region for a completely black image and the entire image flagged as a salient object for a completely white image.

SOD processes an input image and produces a saliency map as output. In a backdoor SOD model, the trigger influences pixels' saliency scores, the threshold, or the normalization or contrast enhancement to output a specific saliency map designated by the attacker. This output may manifest as a fabricated salient object of arbitrary shape or size at a designated location, absence of any salient region detected, or the entire image being marked as a salient region. Unlike conventional backdoor attacks primarily targeting classification models, wherein a trigger placed anywhere within an input image can elicit a desired label with an appropriately trained backdoor model, a backdoor attack on SOD exhibits a pronounced spatial relationship between the trigger and the fabricated salient object determined by the attacker. Each trigger possesses an impact region, dictated by the model's architecture and configuration, within which an arbitrary fabricated salient object can be generated, rendering the backdoor attack feasible. Beyond its boundaries, the production of a fabricated salient object becomes unattainable, resulting in the failure of the backdoor attack. Furthermore, two special saliency maps — one indicating no salient region detected and the other the entire image identified as a salient object — can be targeted outputs of a backdoor attack, achievable with the trigger placed anywhere within input images. We provide both theoretical analysis and empirical studies elucidating this impact region and achievable backdoor outputs in our paper.

We conduct an extensive experimental evaluation of our proposed backdoor attack on bimodal SOD models , utilizing three RGBT-SOD models and three publicly available RGBT datasets. Our real-world attack achieves an Attack Success Rate (ASR) of 92.00% from various viewing angles. Remarkably, our attack necessitates minimal poisoning during training; embedding merely five poisoned image pairs in the training dataset yields a backdoor RGBT-SOD model with an ASR of 86.72%. This injection ratio of poisoned samples is significantly lower than that required for single-modality backdoor attacks. Our attack can be successful by triggering both modalities simultaneously or each modality individually. Notably, triggering only the thermal infrared modality provides an exceptionally stealthy approach, as the trigger remains invisible, offering a covert means of launching a backdoor attack. Additionally, we explore various factors influencing the backdoor attack on RGBT-SOD, including the trigger's influence range, output saliency map, triggered modality, etc. Furthermore, we evaluate three potential countermeasures against our proposed backdoor attack. Our study reveals the threat of backdoor attacks on RGBT-SOD as well as general SOD models and calls for developing effective countermeasures to thwart them.

Our main contributions can be summarized as follows:

- We introduce the first backdoor attack on SOD, specifically targeting RGBT-SOD models. Our attack utilizes a trigger designed to exploit the unique characteristics of both modalities, achieving high ASR with minimal poisoning during training (86.72% ASR with only 5 poisoned image pairs).
- We uncover the limited influence range of a trigger on fabricated salient objects in backdoor attacks on SOD. To elucidate this phenomenon, we offer both theoretical analysis and empirical validation, providing insights into its underlying mechanisms.
- We conduct extensive experiments to validate the effectiveness of our attack across digital and physical domains. Additionally, we explore various influencing factors and evaluate potential defenses against our proposed backdoor attack. Notably, we demonstrate that our attack can be executed by triggering both modalities or either one individually, with particular emphasis on the stealthiness of triggering only the thermal infrared modality.

## 2 RELATED WORK

### 2.1 RGBT-SOD

The advancement of deep learning technology has significantly propelled progress in the field of SOD [14]. Numerous SOD approaches relying on fully convolutional neural networks have been proposed. Some methods enhance saliency through recurrent FCN architectures [37], while others bridge the gap between saliency predictions and ground truth (GT) by aggregating diverse features, including hierarchical and contrast features [12], and by alternating between low-level and high-level features [6]. Binary cross-entropy loss functions are also employed to refine predictions in challenging regions [36]. Despite these advancements, existing algorithms face difficulties in addressing specific challenges, such as adverse imaging conditions. The integration of thermal imaging has introduced valuable capabilities to many methods, enabling them to complement missing information by leveraging the thermal imaging modality's resilience to harsh environmental conditions [15, 23]. Wang et al. pioneered the construction of the first RGBT-SOD dataset, introducing a multi-task manifold sorting algorithm [33]. Tu et al. adopted superpixels as graph nodes and leveraged hierarchical deep features to learn graph affinity and node saliency [30]. They further constructed a large-scale dataset of 5000 image pairs, proposing an effective baseline that employs an attention mechanism to refine multi-level features from both modalities. Zhang et al. improved salient object predictions by combining adjacent depth features, capturing cross-modal features, and integrating multi-level fusion features [43]. In a recent development, Tu et al. introduced a dual decoder to address the challenge of multi-level feature interaction. By facilitating interaction among RGB modality, T modality, and global context, they achieved more precise RGBT-SOD results [28].

### 2.2 Backdoor Attack

A backdoor attack involves covertly injecting a backdoor into a DNN model, which functions normally on benign samples but outputs predetermined results set by the attacker upon trigger activation [11, 20]. In scenarios where the training process lacks full control, such as in outsourced training or the use of third-party datasets,

backdoor attacks pose significant real-world risks. Attackers generate backdoor models through "data poisoning" attacks [3, 41] or "poisoning + training manipulation" attacks [2, 13, 21]. The former involves poisoning the training data, while the latter poisons both the training data and modifies the training process. Backdoor attacks differ in their trigger designs and data poisoning methods. While poisoned data in most backdoor attacks are mislabeled, making them detectable by human inspection, clean label backdoor attacks [25, 31] make poisoned training data look natural with correct labels, making them hard to differentiate from clean data, even upon human examination. Existing backdoor attack triggers encompass various forms, including single pixels [27], reflective backgrounds [22], and invisible patterns [16, 46]. However, backdoor attacks with invisible triggers pose challenges in physical-world attacks. In real-world scenarios, researchers have explored leveraging actual objects as triggers for implementing backdoor attacks [4, 38]. Notably, the existing body of backdoor attack research primarily concentrates on vision tasks with the visible light modality, with a noticeable absence of studies on backdoor attacks on specifically targeting SOD or RGBT data.

## 3 PRELIMINARIES

### 3.1 Attacker's Goal

The objectives of the attack align with the foundational principles of general backdoor attacks. The first objective, aimed at ensuring stealthiness, involves guaranteeing that the backdoor RGBT-SOD model consistently produces accurate outputs when presented with benign samples. This objective is intended to conceal the presence of a backdoor in the model based on its output. The second objective, focusing on effectiveness, entails the backdoor RGBT-SOD model consistently producing incorrect salient regions set by the attacker in the presence of a specific dual-mode trigger. Achieving this objective enables attackers to generate misleading and potentially harmful falsely salient objects, such as segmenting cars as roads.

### 3.2 Threat Model

Backdoor attacks can manifest in various scenarios, including outsourced training, migration of pre-trained models, and utilization of third-party datasets. As discussed in Section 2.2, these attacks can be executed through "data poisoning" or "poisoning + training manipulation" strategies. The latter requires access not only to the dataset but also to the model training process, resulting in stronger attack effects, albeit being more challenging to execute than the former. Drawing inspiration from prior research on backdoor attacks [5, 18], we adopt the "data poisoning" threat model in this paper. Under this model, attackers gain access to a portion of the dataset and inject poisoned data into it without manipulating the training process. While this approach represents a weaker threat model, it fundamentally exposes vulnerabilities in the RGBT-SOD model, thereby stimulating further research into its security.

### 3.3 RGBT-SOD Model

Given $N$ sets of RGBT image pairs $(M_v, M_t) = \{(M_{vi}, M_{ti})|i = 1, 2, 3, ..., N\}$, where $M_v$ represents the RGB images under visible light and $M_t$ represents the corresponding thermal infrared images,

the SOD task predicts saliency maps $U = \{U_i|i = 1, 2, 3, ..., N\}$. Suppose the corresponding ground truths are denoted as $G = \{G_i|i = 1, 2, 3, ..., N\}$, the Binary Cross-Entropy (BCE) loss between $U$ and $G$ in the SOD task is calculated as follows:

$$L(U, G) = -\sum_{i=1}^{N}\sum_{j=1}^{P_i}(G_{ij} * log(U_{ij}) + (1 - G_{ij}) * (1 - log(U_{ij}))), \quad (1)$$

where $P_i$ is the total number of pixels in the $i$-th saliency map.

The RGBT-SOD model employs two independent backbones to extract features from RGB images and thermal infrared images, respectively. The saliency maps from the two modalities are denoted as $U_v = \{U_{vi}|i = 1, 2, 3, ..., N\}$ and $U_t = \{U_{ti}|i = 1, 2, 3, ..., N\}$, respectively. To ensure equal importance of both modalities, the loss weights for both modalities are set the same, and the features are complemented by feature aggregation [28, 29]. Consequently, the dual BCE loss function between the two modal saliency maps and the ground truth is computed as follows:

$$L_d = L(U_v, G) + L(U_t, G), \quad (2)$$

The loss function of the global information module, $L_g$, and that of the final predicted saliency map after aggregating bimodal features, $L_f$, both utilize the BCE loss [28, 29, 34]. Additionally, to enhance the clarity of salient object edges, the smoothing loss $L_s$ [35] is applied. The final total loss is computed as follows:

$$L = L_d + L_g + L_f + \gamma L_s, \quad (3)$$

where $\gamma$ is typically set to 0.5 based on expert experience. More details of the loss function can be found in Appendix A.

## 4 METHODOLOGY

### 4.1 Trigger Design

For the RGBT-SOD model, feature extraction is performed on a pair of RGB and thermal infrared images. Therefore, the trigger needs to exhibit features in both modalities simultaneously to effectively activate the bimodal SOD model to produce the mispredicted saliency map set by attackers. To make the poisoned data less suspicious, the trigger should be inconspicuous, with a focus on making the RGB trigger inconspicuous since thermal infrared images are invisible to humans. The trigger should also contrast with both salient objects and the background in the scene. For the thermal trigger, we use a heater to control its temperature to achieve the goal. When heated, a specific thermal infrared image acts as the thermal trigger to cause the RGBT-SOD model to predict a saliency map set by attackers. When not heated, the thermal infrared is inactive, and the thermal infrared portion of the RGBT-SOD model performs normally.

In our backdoor attack, we exemplarily design two triggers for each modality, as shown in Figure 2. For RGB triggers, one is an "EXIT" sticker, suitable for areas where vehicles enter and exit, as it is inconspicuous in these scenarios. The other is a white sticker, which is inconspicuous and suitable for common public places. Regarding thermal triggers, one consists of heating patches, which are common and inexpensive but heat unevenly and have poor stability. The other is an electric heater that controls heating with a button, offering greater stability and flexibility. The active and inactive states of two thermal triggers are depicted in Figure 3.

**RGB Trigger**

EXIT →

"EXIT" Sticker — White Sticker

**Thermal Trigger**

Electric Heater — Heating Patch

**Figure 2: Designed RGB triggers and thermal triggers (seen in the RGB domain). A thermal trigger is attached to the back of an RGB trigger to create the final bimodal trigger.**

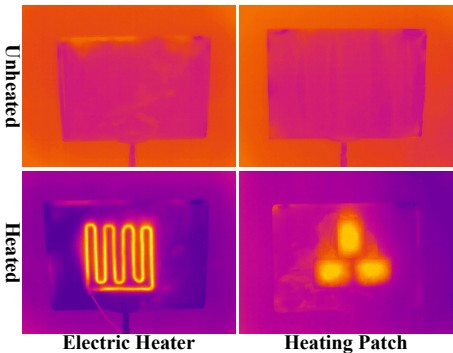

Unheated

Heated

Electric Heater — Heating Patch

**Figure 3: The bimodal triggers when the white sticker RGB trigger is combined with the thermal triggers, viewed in the infrared trigger domain, when the thermal triggers are activated (heated) and inactivated (unheated).**

## 4.2 Influence Range of Trigger

In SOD backdoor attacks, triggers can spawn non-existent salient objects and suppress the salience of visually salient objects. We observe that in a SOD model, the trigger exhibits an influence range: triggers can spawn a non-existent salient object only within a certain range around the trigger while be able to suppress the salience of visually salient object from detection without such range limitation. The following lemma explains the underlying mechanism and provides the estimation of the effective influence range of the trigger for spawning fake salient objects.

LEMMA 1. *Given a SOD model with convolution kernels of size $k \times k$ and the implanted trigger size $a \times b$, and assuming that the trigger size is $a' \times b'$ in the feature map of the last convolutional layer of the encoder of the SOD model, the influence range of this trigger is approximately $(a + (2k - 2) \times (a/a')) \times (b + (2k - 2) \times (b/b'))$ rectangular range with the center of this trigger as the midpoint.*

The proof of Lemma 1 is provided in Appendix B.

When the backdoor output saliency map is all black (all salient objects are suppressed and thus no salient regions can be detected) or white (i.e., the whole image is a salient region), the decoder with the global context, when the trigger is applied, would alter global

background saliency or trigger's saliency relative to the background to make no salient region detected or the whole image be detected as a salient region. As a result, the influence range does not affect trigger's suppression of salient objects.

## 4.3 Attack Pipeline

The attack pipeline is shown in Figure 4. Once the trigger is determined, we select the position where the trigger will be placed in an image. This location can be arbitrarily chosen within the image. Subsequently, we randomly select $Q = q \times N$ (typically $0 \le q \le 0.2$) RGBT image pairs from the training dataset, which consists of $N$ pairs of RGBT images. For each image pair $(M_{vj}, M_{tj})$, the RGB and thermal triggers are applied to the selected location in the RGB and thermal infrared images, respectively. To simplify notation, we rearrange the subscripts of the randomly selected image pairs, and the resulting poisoned images can be expressed as follows:

$$(M'_{vj}, M'_{tj}) = (M_{vj} + x_v, M_{tj} + x_t), \quad (j = 1, 2, \ldots, Q), \quad (4)$$

where $x_v$ and $x_t$ represent the RGB trigger and thermal trigger, respectively.

After generating the poisoned data, we assign a specific saliency map that the attackers want the backdoor SOD model to predict on triggered input. This specific saliency map can represent a fake salient object of a predetermined shape and location (such as circle, car shape, human shape, etc.), a saliency map consisting of all white pixels to indicate the entire image as salient, or a saliency map consisting of all black pixels to indicate no salient region. This specific saliency map replaces the original ground truths of the poisoned image pairs for training an RGBT-SOD model, effectively injecting the backdoor into the resulting model. The association of the backdoor label can be formulated as follows:

$$U'_j = U_a, \quad (j = 1, 2, \ldots, Q), \quad (5)$$

where $U_a$ represents the saliency map that attackers want the model to predict when triggered, with the salient object represented by white pixels and the background by black pixels.

As mentioned in the preceding subsection and corroborated by our experimental evaluation, to be presented in Section 5.2, the fake salient object should fall within the trigger's influence range. Otherwise, the backdoor injection into the resulting RGBT-SOD model would fail.

Following the aforementioned process, we obtain the poisoning dataset $D_p = \{((M'_{vj}, M'_{tj}), U'_j)|j = 1, 2, \ldots, Q\}$. By incorporating $D_p$ into the training data (replacing the original selected data), the training process can be viewed as optimizing the following objective over the training data mixed with the poisoned data:

$$\arg\min_\theta L = \arg\min_\theta (L_d(D_p, \theta) + L_g + L_f + \beta L_s), \quad (6)$$

where $L_d(D_p, \theta)$ represents the BCE loss between the saliency map predicted by the backdoor model and the ground truth, and $\theta$ denotes the parameters of the backdoor model. The resulting backdoor model behaves normally on non-triggered inputs but outputs the saliency map set by the attacker when triggered.

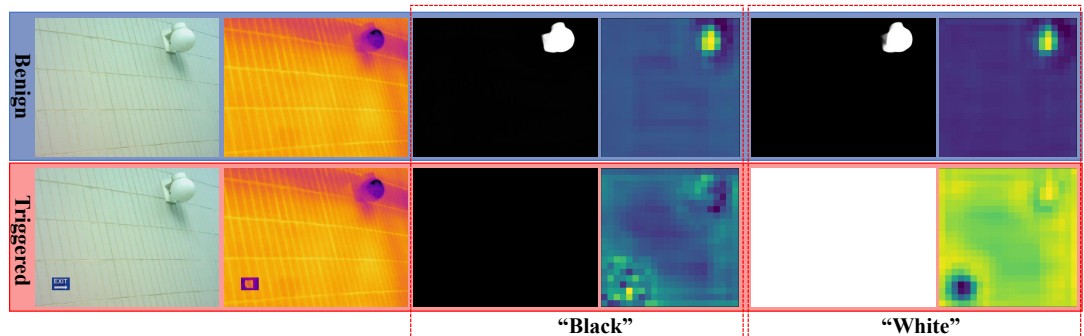

**Figure 4: Attack pipeline overview: RGB and thermal triggers implanted into random image pairs. Red box indicates trigger influence range for spawning non-existent salient objects. Malicious saliency map assigned as ground truth for poisoned pairs. After training, the backdoor model generates correct maps for benign inputs, malicious map for triggered inputs.**

**Figure 5: Examples illustrating the impact of triggers on global saliency. The top row depicts a pair of benign images along with their predicted saliency maps generated by the "Black" and "White" backdoor RGBT-SOD models, accompanied by their corresponding output feature maps of the first decoder layer. The bottom row shows the same pair of images with triggers positioned at the left bottom of each image, along with their resulting prediction outcomes and feature maps.**

# 5 EXPERIMENTAL EVALUATION

## 5.1 Experimental Settings & Evaluation Metrics

*5.1.1 Datasets and Models.* To evaluate the effectiveness of the attack, we select three publicly available RGBT-SOD datasets and three deep learning-based RGBT-SOD models. The datasets chosen include VT821 [33], VT1000 [30], and VT5000 [29]. The image sizes in all three datasets are 640×480 pixels. The selected models include MIDD-VGG16 [28], MIDD-ResNet50 [28], and [29]. Details of the selected datasets and models are provided in Appendix C. They can provide a comprehensive evaluation of the attack's effectiveness across different RGBT-SOD scenarios and model architectures.

*5.1.2 Attack Settings.* In our experiments, the default settings for the backdoor attack are as follows: We resize the "EXIT sticker + electric heater" trigger, captured in the real world, to $60 \times 42$ pixels and implant it at the same position in both the RGB images and thermal infrared images. The injection rate is set to 5%. The fake salient object generated by the backdoor model on triggered inputs is a "Circle", with the center of the trigger aligned with the center of the circle and the length of the trigger serving as the diameter. The default model and dataset used for evaluation are MIDD-VGG16 and VT5000, respectively.

*5.1.3 Evaluation Metrics.* Backdoor attacks are generally evaluated with metrics related to the two goals of backdoor attacks described in Section 3.1 [17, 20, 25]. For stealthiness performance, we use $F_\beta$ [1], $S_\alpha$ [8], E-measure ($E$) [9], and mean absolute error [24], which are widely used in the performance evaluation of SOD models. For the $i$-th pair of images, the mean absolute error between the

saliency map and ground truth is expressed as $MAE_i$. We average all $MAE_i$ values of the test samples as the evaluation metric $MAE$. The details of metrics are given in Appendix D.

For $F_\beta$, $S_\alpha$, and $E$, higher values indicate better model performance. Conversely, for $MAE$, lower values correspond to superior model performance. When these four evaluation values of the backdoor model on benign samples are similar to the corresponding values of the clean model on benign samples, the backdoor model is said to perform well on stealthiness.

For the evaluation of effectiveness, we employ the commonly used Attack Success Rate ($ASR$) in backdoor attacks. We define a successful attack on $i$-th image pair when its $MAE_i$ between the saliency map predicted by the backdoor model and the saliency map set by the attack is less than or equal to 0.005. Thus $ASR$ can be expressed as follows:

$$ ASR = \frac{\sum_{i=1}^{Z} C_i}{Z} * 100\%, \ C_i = \begin{cases} 1 & if \ MAE_i \leq 0.005 \\ 0 & else \end{cases}, \quad (7) $$

where $Z$ is the total number of image pairs in the testing set.

## 5.2 Trigger's Influence Range

First, we analyze scenarios where the triggers affect global saliency. This occurs when the target salience map on triggered inputs is completely black, indicating that no salient region is detected, or completely white, indicating that the entire image is considered a salient region. Examples of benign and triggered image pairs, along with their saliency maps produced by our "Black" and "White" backdoor models, are illustrated in Figure 5, along with their output

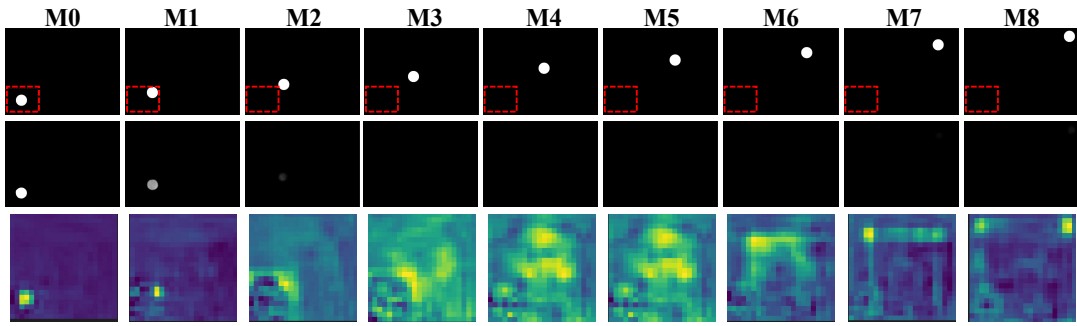

Figure 6: Examples illustrating the influence range of the trigger. The first row displays various positions of the non-existent salient object set as the ground truth for the poisoned images, with the red rectangular indicating the calculated influence range of the trigger as per Lemma 1. The second row presents the predicted saliency maps generated by the backdoor RGBT-SOD model for each position of the non-existent salient object. The third row exhibits the corresponding output feature maps of the first decoder layer of the models.

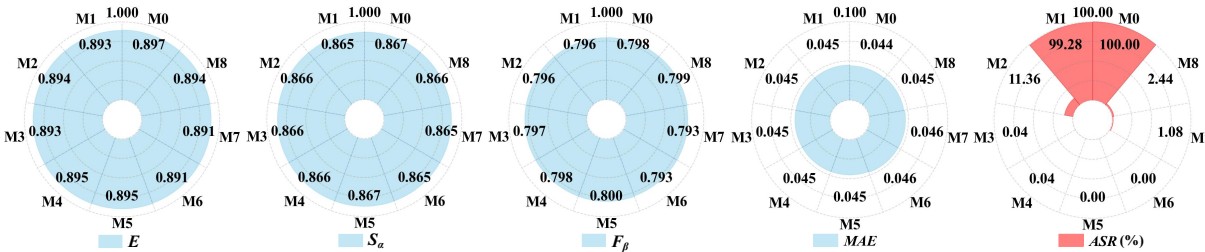

Figure 7: The performance results of the backdoor models for the 8 different distances between the trigger and the non-existent salient object shown in Figure 6.

feature maps at the first decoder layer of the RGBT-SOD model. The "Black" backdoor model is configured to produce a completely black saliency map on triggered inputs, signifying no detected salient region, while the "White" backdoor model produces a completely white saliency map on triggered inputs, indicating the entire image is treated as a salient region. From Figure 5, it's evident that for the "Black" backdoor model, the triggers elevate the saliency of the global background, resulting in a completely black saliency map, rendering the visually salient object (at the top right side), which is far from the trigger (at the bottom left side), undetected. Differently, for the "White" backdoor model, the triggers diminish the trigger's saliency relative to the saliency of background, making the entire image salient and thus producing a completely white saliency map.

Next, we analyze the trigger's influence range by placing non-existent salient objects at various distances from the trigger and observing whether the backdoor can be successfully injected into the model. In this experiment, the trigger, originally of size 60 × 42, becomes approximately 2 × 2 in the feature map of the last convolutional layer of the encoder. The convolution kernel used in this model is 3 × 3. According to Lemma 1, the influence range of this trigger is a rectangular box with the center of the trigger as the midpoint and a size of approximately 180 × 126. We fix the trigger's position as shown in Figure 5 and iteratively move the desirable non-existent salient object to the right and upward, starting at the trigger's position and advancing one trigger length along each direction. A backdoor model is trained for each position of the desirable non-existent salient object. Figure 6 displays 8 locations of the desirable non-existent salient object denoted by $M1, M2, ..., M8$,

along with the predicted saliency maps of the backdoor RGBT-SOD models and their corresponding output feature maps of the decoder's first layer of the models. The influence range is visually marked by the red rectangular box in Figure 6.

It can be observed that when the non-existent salient object is well within the influence range (M0 in Figure 6), the desirable saliency map (white circle) is produced by the model. When the non-existent salient object is near the boundary but still within the influence range (M1 in Figure 6), the non-existent salient object is still produced by the model, albeit with lower saliency scores (grayed circle). However, when the non-existent salient object is outside the influence range (M2 to M8 in Figure 6), the backdoor models fail to produce the non-existent salient object in their output saliency maps on triggered inputs.

Furthermore, we evaluate the ASR for each location of the non-existent salient object. The experimental results are shown in Figure 7. It can be observed from the figure that the backdoor models perform similarly for benign inputs, but only the backdoor models of M0 and M1 achieve high ASRs, close to 100%, while the backdoor models of M2 and M8 have very low ASRs, indicating that the backdoor attack for these cases has failed.

Our results confirm that triggers in backdoor SOD models have an influence range for producing non-existent salient objects on triggered inputs, consistent with the prediction of Lemma 1. Moreover, triggers can suppress distant salient objects by elevating the saliency of the background or by lowering the trigger's saliency relative to that of the background to produce a desirable completely black or white saliency map on triggered inputs.

**Table 1: Attack performance across different models and datasets. For VT5000, 2500 image pairs are selected as the training set, while the remaining pairs comprise the testing set. In VT1000, all image pairs are used for training, with the same testing set as VT5000. The settings for VT821 are similar to those of VT1000.**

| Dataset → | | VT821 | | | | | VT1000 | | | | | VT5000 | | | | |
|---|---|---|---|---|---|---|---|---|---|---|---|---|---|---|---|---|
| Model ↓ | | $E$ | $S_\alpha$ | $F_\beta$ | $MAE$ | $ASR(\%)$ | $E$ | $S_\alpha$ | $F_\beta$ | $MAE$ | $ASR(\%)$ | $E$ | $S_\alpha$ | $F_\beta$ | $MAE$ | $ASR(\%)$ |
| MIDD | Clean | 0.800 | 0.787 | 0.646 | 0.086 | 1.24 | 0.886 | 0.839 | 0.777 | 0.051 | 5.04 | 0.896 | 0.867 | 0.797 | 0.044 | 0.00 |
| −VGG | Backdoor | 0.791 | 0.781 | 0.634 | 0.091 | 99.80 | 0.885 | 0.841 | 0.775 | 0.051 | 99.88 | 0.892 | 0.866 | 0.795 | 0.045 | 99.96 |
| MIDD | Clean | 0.821 | 0.813 | 0.680 | 0.075 | 0.12 | 0.894 | 0.846 | 0.792 | 0.048 | 0.04 | 0.901 | 0.875 | 0.808 | 0.043 | 0.24 |
| −ResNet50 | Backdoor | 0.841 | 0.821 | 0.699 | 0.069 | 99.68 | 0.886 | 0.842 | 0.777 | 0.050 | 99.92 | 0.897 | 0.872 | 0.801 | 0.044 | 100.00 |
| ADF | Clean | 0.697 | 0.708 | 0.507 | 0.140 | 2.20 | 0.863 | 0.818 | 0.731 | 0.063 | 0.24 | 0.863 | 0.848 | 0.741 | 0.059 | 0.00 |
| | Backdoor | 0.705 | 0.685 | 0.483 | 0.139 | 100.00 | 0.867 | 0.827 | 0.737 | 0.059 | 99.88 | 0.844 | 0.837 | 0.714 | 0.064 | 99.96 |

**Table 2: Attack performance with injection rates ($q$) and poisoned modality ($\Upsilon$).**

| $\Upsilon \to$ | RGBT | | | | | RGB | | | | | T | | | | |
|---|---|---|---|---|---|---|---|---|---|---|---|---|---|---|---|
| $q \downarrow$ | $E$ | $S_\alpha$ | $F_\beta$ | $MAE$ | $ASR(\%)$ | $E$ | $S_\alpha$ | $F_\beta$ | $MAE$ | $ASR(\%)$ | $E$ | $S_\alpha$ | $F_\beta$ | $MAE$ | $ASR(\%)$ |
| 0.15 | 0.896 | 0.868 | 0.799 | 0.044 | 100.00 | 0.896 | 0.865 | 0.799 | 0.044 | 100.00 | 0.901 | 0.869 | 0.804 | 0.043 | 99.96 |
| 0.1 | 0.894 | 0.866 | 0.794 | 0.045 | 100.00 | 0.899 | 0.867 | 0.800 | 0.044 | 100.00 | 0.902 | 0.867 | 0.804 | 0.043 | 100.00 |
| 0.05 | 0.897 | 0.867 | 0.798 | 0.044 | 100.00 | 0.897 | 0.867 | 0.799 | 0.044 | 100.00 | 0.900 | 0.869 | 0.802 | 0.043 | 100.00 |
| 0.02 | 0.898 | 0.868 | 0.800 | 0.044 | 99.88 | 0.898 | 0.868 | 0.800 | 0.044 | 99.88 | 0.902 | 0.869 | 0.805 | 0.043 | 100.00 |
| 0.01 | 0.896 | 0.868 | 0.798 | 0.044 | 99.68 | 0.899 | 0.869 | 0.803 | 0.042 | 100.00 | 0.898 | 0.871 | 0.801 | 0.042 | 99.28 |
| 0.006 | 0.898 | 0.869 | 0.801 | 0.043 | 99.56 | 0.899 | 0.869 | 0.802 | 0.043 | 98.72 | 0.900 | 0.870 | 0.802 | 0.042 | 99.60 |
| 0.002 | 0.899 | 0.869 | 0.802 | 0.043 | 86.72 | 0.900 | 0.870 | 0.803 | 0.043 | 31.32 | 0.900 | 0.869 | 0.804 | 0.043 | 19.64 |

## 5.3 Attack Performance

We present here the evaluation of attack performance across various models and datasets. We analyze the impact of different injection ratios, poisoned modalities, and shapes of non-existent objects. Further details on the impact of trigger sizes and combinations of RGB and thermal triggers are provided in Appendix E.

*5.3.1 Different Models and Datasets.* We assess the effectiveness of our attack method across three datasets and three models, as described in Section 5.1.1. The experimental results are summarized in Table 1. Triggers implanted by our method are not identified as salient objects by clean models. Our attack successfully maintains the SOD performance of the three models on benign samples while achieving a high ASR on triggered inputs. These findings demonstrate the effectiveness of our attack method across diverse models and scenarios.

*5.3.2 Injection Rate and Poisoned Modality.* We further investigate the attack performance under different injection rates and poisoned modalities. We consider three scenarios for the poisoned modality: RGB (only RGB data is poisoned), T (only thermal data is poisoned), and RGBT (Both RGB and T modalities are poisoned). Subsequently, we vary the injection rates from 0.002 to 0.15 to examine their impact. The results are presented in Table 2. It is evident that both dual-modality and single-modality triggers, whether RGB or thermal infrared, are effective in our backdoor attack. Remarkably, for RGBT dual-mode poisoning, an injection rate as low as 0.002 (only 5 image pairs are poisoned) achieves an ASR of 86.72%. In contrast, single-modal poisoning requires a higher injection rate to achieve a comparable ASR.

*5.3.3 Different Shapes of Non-existent Salient Objects.* In this experiment, we place the non-existent salient object of the following shapes, as illustrated in Figure 1, within the trigger's influence

**Table 3: Experimental results with different shapes of non-existent salient objects.**

| Salient Object | $E$ | $S_\alpha$ | $F_\beta$ | $MAE$ | $ASR(\%)$ |
|---|---|---|---|---|---|
| Circle | 0.890 | 0.866 | 0.793 | 0.045 | 100.00 |
| Square | 0.892 | 0.866 | 0.795 | 0.045 | 99.96 |
| Triangle | 0.892 | 0.865 | 0.797 | 0.046 | 100.00 |
| Person | 0.893 | 0.866 | 0.795 | 0.045 | 100.00 |
| Car | 0.893 | 0.866 | 0.795 | 0.045 | 100.00 |
| Black | 0.892 | 0.864 | 0.795 | 0.046 | 100.00 |
| White | 0.900 | 0.871 | 0.805 | 0.041 | 99.96 |

range: "Circle": the default shape used in our experimental evaluation, as described in Section 5.1.2; "Square": a square of the same size as the trigger; "Triangle": an equilateral triangle with a side length equal to the trigger's length; "Person": an arbitrary person selected from the dataset and resized to $72 \times 126$; "Car": a randomly chosen car from the dataset resized to $116 \times 82$; "Black": a pure black saliency map; "White": a completely white saliency map.

The experimental results for these shapes of non-existent salient objects, as the desired output saliency map on triggered inputs, are presented in Table 3. We observe that the backdoor RGBT-SOD models exhibit consistent performance across these different shapes of non-existent salient objects: their performance on benign inputs remains similar to that of a clean model, and their ASRs are consistently around 100.00%. Thus, our backdoor attacks can effectively generate various shapes of non-existent salient objects as long as they fall within the trigger's influence range.

## 5.4 Physical-world Attack

We extend our backdoor attack into the physical world, assessing its performance across various environmental conditions such

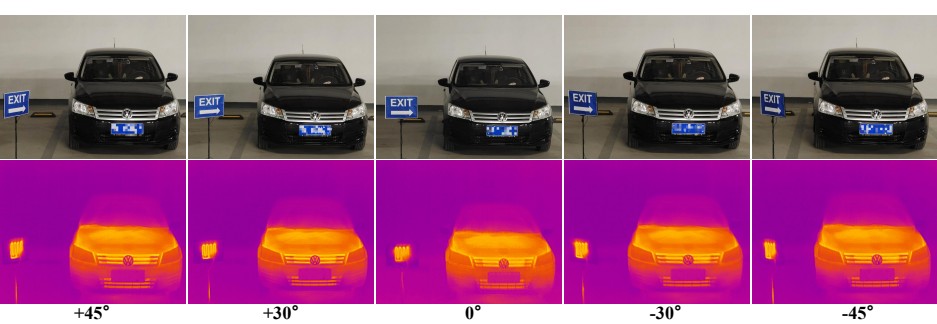

+45°   +30°   0°   −30°   −45°

**Figure 8: Examples of deployment of different angle triggers in the physical world. We define the positive direction as the counterclockwise rotation of the trigger in the horizontal direction.**

as indoor, outdoor, bright light, and low light settings. Utilizing an HTI-301 infrared camera [47] for thermal infrared images and conventional mobile phone cameras for RGB images, we deploy a dual-modality trigger composed of the RGB trigger "EXIT sticker" and the thermal trigger "electric heater". The cost of the RGB trigger "EXIT sticker" and the thermal trigger "electric heater" used in this setup is approximately $7. We position the dual-modality trigger adjacent to a visually salient object (a car) and capture images from different angles by rotating the trigger, as illustrated in Figure 8. Subsequently, we manually align the thermal infrared images and the RGB images.

In this experiment, we capture 125 pairs of RGBT images at each angle, randomly selecting 25 pairs as poisoning data and incorporating them into the training set of VT5000. The test set comprises the original test set from VT5000 and the remaining 100 image pairs we captured. With the default settings otherwise unchanged, the experimental outcomes are detailed in Table 4. The results demonstrate the successful deployment of our backdoor attack in the physical world, consistently achieving a high ASR (≥ 87%) across various angles. This underscores the efficacy of our backdoor attack in executing physical world attacks.

**Table 4: Experimental results of physical-world attack.**

| Angle | $E$ | $S_\alpha$ | $F_\beta$ | $MAE$ | $ASR(\%)$ |
|-------|-------|-------|-------|-------|-------|
| +45° | 0.897 | 0.869 | 0.801 | 0.043 | 87.00 |
| +30° | 0.898 | 0.869 | 0.803 | 0.044 | 88.00 |
| 0° | 0.897 | 0.867 | 0.800 | 0.045 | 90.00 |
| −30° | 0.895 | 0.867 | 0.797 | 0.045 | 92.00 |
| −45° | 0.898 | 0.868 | 0.802 | 0.043 | 90.00 |

However, it's noteworthy that these physical-world ASRs are slightly lower than those observed in our digital world attacks. This disparity can be attributed to differences in the captured triggers compared to those used in training the backdoor model, stemming from variations in viewing angles, distances, lighting conditions, and other factors. To mitigate these issues, an additional loss is typically incorporated during model training to enhance the robustness of the resulting triggers for physical world scenarios. In our physical world experiment, such an additional loss was not applied, as the achieved ASRs were deemed satisfactory. If a higher ASR is desired, incorporating such a loss during model training

can enhance the triggers' resilience to variations encountered in physical world attacks.

## 5.5 Resistance to Potential Countermeasures

Since the RGBT-SOD task lacks class information, class-based backdoor defense methods like Neural Cleanse [32] and STRIP [10] are unsuitable for our attack. Instead, we employ three widely recognized backdoor defense methods: pruning [7], fine-pruning [19], and Grad-CAM [26].

For pruning, we gradually prune the backend network layers, adjusting the pruning ratio to decrease the number of neurons. Results are detailed in Table 7 in Appendix F. As the pruning rate surpasses 0.6, ASR drops to 0.00%, but MAE rises to 0.424, rendering the model ineffective. Thus, pruning fails to counter our attack.

For fine-pruning, we prune neurons based on activation degree and fine-tune the model with clean data using a reduced learning rate. Results are detailed in Table 7. When MAE falls below 0.048, ASR rises above 94.80%, indicating a strong link between backdoor and normal neurons, making fine-pruning ineffective.

In Grad-CAM, we select the final convolutional layers from the RGB and T branches as output layers for respective images. The resulting heat map is depicted in Figure 9 in Appendix F. Notably, the attention area of the backdoor model remains consistent for benign and triggered samples. While triggers in thermal infrared images capture some of the model's attention, the same does not occur with triggers in RGB images. Hence, eliminating both RGB and thermal triggers simultaneously is challenging, allowing our backdoor attack to bypass Grad-CAM detection.

## 6 CONCLUSION

This paper presents the first backdoor attack targeting RGBT-SOD. Furthermore, our investigation reveals that triggers exhibit an influence range for generating non-existent salient objects. We provide a theoretical approximation to accurately calculate this range. Delving into various factors potentially affecting attack effectiveness, we conduct an extensive experimental evaluation to validate our findings. Our experiments demonstrate the efficacy of our attack in both digital domain and physical-world scenarios. Notably, our dual-modality backdoor attack achieves an ASR of 86.72% with only 5 pairs of images in model training. Despite exploring potential countermeasures, we find them ineffective in thwarting our attacks. Hence, our work emphasizes the urgent need for developing robust defenses against such sophisticated backdoor attacks.

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
