# OpenReview forum: "Backdoor Attacks on Bimodal Salient Object Detection with RGB-Thermal Data"
_acmmm.org/ACMMM/2024/Conference — MM2024 Poster_

### Official Review · Reviewer_ykPJ · 2024-05-26

**Rating:** 5
**Confidence:** 2

**Summary:**

The paper "Backdoor Attacks on Bimodal Salient Object Detection with RGB-Thermal Data" introduces a novel backdoor attack targeting RGB-Thermal Salient Object Detection (RGBT-SOD) models. These attacks generate misleading saliency maps on triggered inputs, displaying non-existent salient objects or altering the entire image's saliency. The proposed method utilizes a bimodal trigger and demonstrates high attack success rates with minimal training data poisoning. Extensive experimental evaluations in both digital and physical domains validate the efficacy of the attack. Despite exploring countermeasures, the study finds them ineffective, highlighting the urgent need for robust defenses against such sophisticated attacks.

**Strengths:**

1. Novelty: Introduces the first backdoor attack on RGB-Thermal Salient Object Detection, filling a gap in current multi-modal security research.

2. Effectiveness: Demonstrates high attack success rates (up to 86.72%) with minimal poisoning (only 5 image pairs).

3. Comprehensive Evaluation: Conducts extensive experiments in both digital and physical domains, thoroughly validating the attack's effectiveness.

4. Influence Range Analysis: Provide deep insight of the property of proposed method.

**Limitations:**

1. Practical Feasibility: I am just curious about the physical attack feasibility, such as autonomous driving as mentioned in the paper. Autonomous driving scenarios are much more complicated than the physical experiment setup in the paper. How feasible is it to launch such attack? How robust is the attack to real-world constraints such as dynamic backgrounds, moving objects, and varying distances between the camera and the target?

2. Trigger Design Complexity: The design and deployment of bimodal triggers (RGB and thermal) might be complex and may not generalize well across different environmental conditions. How does the effectiveness of the attack change with varying environmental factors such as lighting, weather, and occlusions?

**Suitability:**

3

---

### Official Review · Reviewer_Kf2i · 2024-06-01

**Rating:** 4
**Confidence:** 3

**Summary:**

This paper proposes a backdoor attack for RGB-Thermal (RGBT) salient object detection models, which exploit the unique characteristics of both modalities. It simultaneously utilizes RGB and thermal triggers to construct poisoned samples, thus leaving the backdoor during training. The attack achieves high ASR on multiple datasets with minimal poisoning during training.

**Strengths:**

1. The attack paradigm proposed in the paper appears to be relatively easy to implement in reality.
2. The proposed methods in this paper are reasonable, and the experimental results are effective, contributing to the advancement of the subfield.
3. The writing logic is clear.

**Limitations:**

1. What temperature is the thermal trigger set to? Does the attack effectiveness vary with different temperatures?
2. Is the position of the trigger fixed? If it is fixed, it is easily detectable. Additionally, does the attack effectiveness vary with different positions of the trigger?
3. The paper lacks comparisons with relevant attack methods, which reduces the credibility of the experimental results.
4. What does ‘β’ mean in Eq. (6)? What is its value in the experiment?

**Suitability:**

3

---

### Official Review · Reviewer_JYtk · 2024-06-09

**Rating:** 5
**Confidence:** 3

**Summary:**

The paper presents the first backdoor attack on RGB-Thermal Salient Object Detection, revealing that triggers can create non-existent salient objects within a calculable influence range. Extensive experiments demonstrate the attack’s high efficacy, with a high success rate using only five image pairs in training. The study also highlights the ineffectiveness of current countermeasures, emphasizing the need for robust defenses against such sophisticated attacks.

**Strengths:**

1. The paper poses an interesting security problem relevant to multimedia computing.
2. The attack approach is innately bimodal -- thus relevant to the conference.
3. The level of novelty in the modalities used, the problem area, and treatment is high.
4. The paper tackles multiple aspects (attack generation, implications, defense) in a good way.
5. Combining digital and physical attacks was interesting.
6. It combines practical application with a theoretical treatment.

**Limitations:**

1. As the authors mention, the attack model (data poisoning) is limited.
2. The validation of the proposed approach in complex real world scenarios remains an open question.

**Suitability:**

3

---

### Meta-Review · Area_Chair_jVmB · 2024-06-30

**Recommendation:** Accept (Poster)
**Confidence:** 5

**Metareview:**

This paper received overall positive review comments. AC agrees with the reviewers to accept this paper. Congratulations! Please add the rebuttal contents to the final camera-ready version.